# Outlier Detection for Mammograms

**Ryan Zurrin**                    RYAN.ZURRIN001@UMB.EDU
**Neha Goyal**                     NEHA.GOYAL001@UMB.EDU
**Pablo Bendiksen**                P.BENDIKSEN001@UMB.EDU
**Muskaan Manocha**                MUSKAAN.MANOCHA001@UMB.EDU
**Dan Simovici**                   DAN.SIMOVICI@UMB.EDU
**Nurit Haspel**                   NURIT.HASPEL@UMB.EDU
**Marc Pomplun**                   MARC.POMPLUN@UMB.EDU
**Daniel Haehn**                   DANIEL.HAEHN@UMB.EDU

## Abstract

Mammograms are vital for detecting breast cancer, the most common cancer among women in the US. However, low-quality scans and imaging artifacts can compromise their efficacy. We introduce an automated pipeline to filter low-quality mammograms from large datasets. Our initial dataset of $176,492$ mammograms contained an estimated $5.5\%$ lower quality scans with issues like metal coil frames, wire clamps, and breast implants. Manually removing these images is tedious and error-prone. Our two-stage process first uses threshold-based 5-bin histogram filtering to eliminate undesirable images, followed by a variational autoencoder to remove remaining low-quality scans. Our method detects such scans with an F1 Score of $0.8862$ and preserves $163,568$ high-quality mammograms. We provide results and tools publicly available as open-source software.

**Keywords:** anomaly detection, outlier detection, mammograms, unsupervised learning

## 1. Introduction

Breast cancer, a prevalent cause of death among women (Yusuf et al., 2021; Lei et al., 2021), can be better managed with early detection and advanced machine-learning tools (Lotter et al., 2021). For a robust machine learning classifier, one strategy is to unify the quality and content of training data by removing low-quality images and outliers (Chandola et al., 2009; Smiti, 2020; Shvetsova et al., 2021). We are building an extensive, publicly available mammography database from which we began with $967,991$ mammograms acquired by our collaborators. Through data cleaning using metadata such as small dimensions and manufacturer, we reduced the number of images to $176,492$ mammograms, but an estimated $5.5\%$ remained low-quality. Manually selecting these images would be infeasible, prompting us to evaluate 26 unsupervised outlier detection algorithms, including traditional and deep learning-based approaches (Section 2). Based on various experiments, we introduce 5-BHIST, a thresholded histogram-binning method paired with a variational autoencoder. This two-stage outlier detection pipeline significantly outperforms other unsupervised machine learning algorithms in detecting low-quality mammograms.

## 2. Experimental Setup

***Test Datasets.*** We initially created the five representative test datasets A, B, C, A\*, and B\* with varying proportions of unwanted images (between 5 and 24%) by randomly sampling

Figure 1: **Two-stage Outlier Detection.** Our method combines a 5-bin histogram filtration technique (5-BHIST) with a variational autoencoder (VAE) to automatically eliminate undesirable images. We perform experiments on a total of 6 test datasets from our initial collection of mammograms. With optimized parameters and normalization methods, we reduce the amount of low-quality mammograms by 83.15%.

100 and 1000 mammograms from our original collection. We manually selected undesired images through multiple consensus-driven user studies with 9 participants. After our initial experiments, we filtered our large collection of mammograms with the best approach (5-BHIST). We then randomly sampled dataset C* for additional testing to identify the optimal algorithm for the second filtering stage (Figure 1).

***Normalization.*** We applied various normalization methods to ensure comparable pixel intensities across different device manufacturers (Patro and Sahu, 2015). *Max:* Re-scale intensities between -1 and 1: $x_{\text{scaled}} = x/max(|x|)$. *Min-Max:* Re-scale intensities to the fixed range [0, 1]: $x_{\text{norm}} = (x_{\text{i}}-x_{\text{min}})/(x_{\text{max}}-x_{\text{min}})$. *Gaussian:* Introduce a blur: $x_{\text{gaussian}} = (x_{\text{gaussian\_filter(sigma=20)}})/x_{\text{max}}$. *zscore:* Standardize across a normal distribution: $x_{\text{gaussian}} = (x_i - \mu)/\sigma$. *Robust:* Scale data using median subtraction and IQR division: $x_{\text{robust}} = (x_i - \mu)/(IQR)$.

***Image Features.*** We utilize image feature descriptors to reduce the number of data points per mammogram. *Full-intensity histograms*, with bin sizes selected automatically based on pixel ranges; *Downsampling*, which reduces the spatial resolution via stretching (without anti-aliasing); *Scale-invariant feature transforms (SIFT)*, used to create keypoints (Lowe, 2004); *Oriented FAST and rotated BRIEF (ORB)*, similar to *SIFT* (Rublee et al., 2011).

***Algorithms.*** We carried out unsupervised outlier detection on all our test datasets using 26 distinct algorithms from the PyOD[1] software package (Zhao et al., 2019, 2021; Han et al., 2022), with a total number of 340 configured experiments across all tests.

***Evaluation Metric.*** To quantify outlier detection success, we measure the F1 Score as *F1 = 2 \* (precision \* recall) / (precision + recall)* (Powers, 2011).

## 3. Results

We fully tuned the 26 anomaly detection algorithms available in PyOD for comparative analysis and evaluated the best-performing configurations on our representative test datasets (Table 1). Initial results indicated a preference for a specific normalization and feature descriptor configuration: Histogram binning after Gaussian normalization. We then performed ablation studies regarding the number of histogram bins. We compared different

---

1. Python Outlier Detection (PyOD) available at https://github.com/yzhao062/pyod

Table 1: **Outlier Detection Results.** Utilizing best-performing normalization and features (G: Gaussian, M: Max, MM: Min-Max, R: Robust, Z: Z-Score, H: Histogram, S: SIFT, O: ORB), our 5-BHIST method yields the highest average F1 Score of 0.8772 on varied test datasets. Incorporating a variational autoencoder (VAE) as a second-stage algorithm elevates this to 0.8862.

| Algorithm | A (n=100, 8%) | | B (n=100, 13%) | | C (n=100, 24%) | | A* (n=1000, 6.3%) | | B* (n=1000, 5.0%) | | C* (n=1000, 1.5%) | |
|---|---|---|---|---|---|---|---|---|---|---|---|---|
| AE | M + H | 0.2500 | M + S | 0.3077 | M + S | $0.4917 \pm 0.0167$ | M + H | 0.1270 | M + H | 0.1200 | MM + H | 0.1391 |
| AvgKNN | G + H | 0.6250 | G + H | 0.6923 | G + H | 0.8333 | G + H | 0.7460 | G + H | 0.6600 | Z + S | 0.0522 |
| VAE | MM + H | 0.2500 | M + S | 0.3077 | MM + S | $0.6000 \pm 0.0333$ | MM + H | 0.1111 | MM + H | $0.0940 \pm 0.0092$ | **MM + H** | **0.1530±0.0070** |
| SOGAAL | M + S | $0.0250 \pm 0.0500$ | G + O | 0.0000 | M + H | 0.0000 | M + S | 0.0000 | M + S | 0.0000 | M + S | $0.0124 \pm 0.0247$ |
| DeepSVDD | G + H | $0.6750 \pm 0.0612$ | G + H | $0.6978 \pm 0.0111$ | G + H | $0.2913 \pm 0.2867$ | G + H | $0.5322 \pm 0.1676$ | G + H | $0.4292 \pm 0.0619$ | MM + H | $0.1009 \pm 0.0403$ |
| AnoGAN | G + H | 0.0000 | M + S | 0.0769 | M + O | 0.2083 | G + H | 0.0000 | G + H | 0.0000 | Z + S | 0.1043 |
| HBOS | G + H | 0.6250 | M + H | 0.4615 | G + H | 0.8261 | G + H | 0.7885 | G + H | 0.7805 | M + H | 0.1217 |
| LOF | MM + S | $0.1750 \pm 0.0612$ | MM + S | 0.3077 | MM + S | $0.6000 \pm 0.0333$ | MM + S | $0.5095 \pm 0.0321$ | MM + S | $0.6100 \pm 0.0257$ | M + S | 0.1391 |
| OCSVM | G + H | 0.0000 | G + H | 0.0000 | G + H | 0.0000 | G + H | 0.0000 | G + H | 0.0000 | G + O | 0.0696 |
| IForest | G + H | 0.5000 | G + H | 0.6154 | G + H | 0.5833 | G + H | $0.6739 \pm 0.0328$ | G + H | $0.6473 \pm 0.0101$ | M + H | $0.1148 \pm 0.0260$ |
| CBLOF | G + H | 0.6250 | G + H | 0.6923 | G + H | 0.8333 | G + H | $0.7492 \pm 0.0063$ | G + H | 0.0202 | Z + S | $0.0452 \pm 0.0085$ |
| COPOD | G + H | 0.3750 | G + H | 0.3846 | G + H | 0.6250 | G + H | 0.3651 | G + H | 0.4583 | R + H | 0.1217 |
| SOS | M + S | $0.4750 \pm 0.0500$ | M + S | $0.5385 \pm 0.0973$ | MM + S | $0.7167 \pm 0.0312$ | M + S | $0.2159 \pm 0.0384$ | M + S | $0.5240 \pm 0.0265$ | M + S | 0.1217 |
| KDE | G + H | 0.0000 | G + H | 0.0000 | G + H | 0.0000 | G + H | 0.0000 | G + H | 0.0000 | M + O | 0.0000 |
| Sampling | G + H | $0.5750 \pm 0.0612$ | G + H | $0.5077 \pm 0.0377$ | G + H | $0.6500 \pm 0.1007$ | G + H | $0.5508 \pm 0.2622$ | G + H | $0.3341 \pm 0.3221$ | Z + S | $0.0417 \pm 0.0085$ |
| PCA | G + H | 0.3750 | G + H | 0.4800 | G + H | 0.5366 | G + H | 0.3651 | G + H | 0.4783 | MM + H | 0.1391 |
| LMDD | G + H | 0.0000 | M + S | $0.1692 \pm 0.0897$ | MM + S | $0.2250 \pm 0.1225$ | G + H | 0.0000 | G + H | 0.0000 | M + O | 0.1217 |
| COF | G + H | 0.6250 | MM + S | 0.3077 | M + S | 0.6250 | G + H | 0.1746 | G + H | 0.1000 | M + S | 0.1217 |
| ECOD | G + H | 0.5333 | G + H | 0.6154 | G + H | 0.6250 | G + H | 0.7097 | G + H | 0.6600 | R + H | 0.1217 |
| KNN | G + H | 0.6250 | G + H | 0.6400 | G + H | 0.8085 | G + H | 0.7460 | G + H | 0.6600 | M + S | 0.0522 |
| MedKNN | G + H | 0.6250 | G + H | 0.6923 | G + H | 0.8333 | G + H | 0.7460 | G + H | 0.6600 | Z + S | 0.0522 |
| SOD | MM + S | $0.3500 \pm 0.0935$ | MM + S | $0.4308 \pm 0.0615$ | MM + S | $0.6167 \pm 0.0167$ | MM + S | $0.2714 \pm 0.0404$ | MM + S | $0.2000 \pm 0.0346$ | MM + S | 0.0870 |
| INNE | M + S | $0.5500 \pm 0.0612$ | MM + S | $0.6308 \pm 0.0308$ | MM + S | $0.7833 \pm 0.0312$ | M + S | $0.3444 \pm 0.0471$ | M + S | $0.4280 \pm 0.0431$ | MM + S | $0.1530 \pm 0.0170$ |
| FB | M + S | 0.2500 | G + H | 0.3077 | G + H | 0.6250 | M + S | $0.4476 \pm 0.0525$ | M + S | $0.5900 \pm 0.0392$ | MM + S | $0.1496 \pm 0.0085$ |
| LODA | G + H | $0.3800 \pm 0.1122$ | G + H | $0.4017 \pm 0.1585$ | G + H | 0.4167 | G + H | $0.3312 \pm 0.3241$ | G + H | $0.5019 \pm 0.3246$ | Z + H | $0.0522 \pm 0.0156$ |
| SUOD | G + H | 0.5000 | G + H | $0.5742 \pm 0.0444$ | G + H | $0.6583 \pm 0.0408$ | G + H | $0.6926 \pm 0.0104$ | G + H | $0.6446 \pm 0.0079$ | M + H | $0.0939 \pm 0.0085$ |
| **5-BHIST** | **G + H** | **0.8571** | **G + H** | **0.8696** | **G + H** | **0.9333** | **G + H** | **0.8908** | **G + H** | **0.8352** | N/A | N/A |

bin configurations ($b = 2, 5, 10$), optional Gaussian blur with varying sigma ($\sigma = 5, 10, 20$), and all normalization techniques with a 2-bin limitation based on previous explorations. Min-max normalization outperformed Gaussian, highlighting bin size as a critical factor for optimal algorithm performance. However, a 2-bin approach contributed to significant false positive classifications. Further ablation studies and consensus-driven inspection confirmed a setting of 5-bins with a bi-conditional thresholding operation (bins $b_2 < 2000$ and $b_5 > 15,000$) for high F1 scores. We report the performance of 5-BHIST in Table 1.

**Limitations.** Our evaluations are based on algorithm tuning from representative mammogram subsets and validated by user studies; thus, results are estimates. Future public access to our full mammogram collection will allow broader expert validation.

## 4. Conclusions

We evaluate 26 unsupervised algorithms for filtering low-quality mammograms in extensive data collections. Our findings indicate that a combination of min-max normalized histogram binning paired with a variational autoencoder can detect unwanted images with an average F1 Score of 0.8862. This reduces the number of unwanted images in our collection by 5.93x, from an estimated 9,708 low-quality scans to 1,636. Our final dataset now contains 1% unwanted images as validated by manual inspection. All code, data, experiments, and additional information are available at *https://github.com/mpsych/ODM*.

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
