# OpenReview forum: "Outlier Detection for Mammograms"
_MIDL.io/2023/Short_Paper_Track — MIDL 2023 Short paper track Poster_

### Official Review · Reviewer_Aruq · 2023-04-21

**Rating:** 6
**Confidence:** 4

**Review:**

This paper proposes an automated pipeline to filter low-quality mammograms from large datasets. The paper is generally well written with the methods being easy to understand and follow. However, there are some minor issues needed to be addressed in the revision:
1. The authors claim it is a publicly available dataset. The release link should be provided.
2. Many parameters are utilized in method. The parameter analysis should be provided.

---

### Official Review · Reviewer_DJjf · 2023-04-24
**Issues include non-robust normalization, insufficient 5-bin histogram explanation, and feasibility of discarding images with small artifacts**

**Rating:** 5
**Confidence:** 4

**Review:**

The paper's main contribution is the development of an automated pipeline that effectively filters low-quality mammograms from large datasets. This is achieved through a two-stage process involving threshold-based 5-bin histogram filtering and a variational autoencoder, which removes undesirable images, such as those with metal coil frames, wire clamps, and breast implants. The method achieves an F1 score of 0.8862 in detecting low-quality scans and preserves 163,568 high-quality mammograms. The authors provide open-source software for public use.

However, the paper could benefit from addressing some issues. Firstly, the min-max intensity normalization method used may not be robust to intensity outliers. Additionally, the 5-bin histogram method could be explained more thoroughly for readers to understand how it works. Finally, in clinical applications, it may not be feasible to discard all images with small artifacts. The paper could discuss possible solutions for this issue.